# Role of Autophagy in Proteostasis: Friend and Foe in Cardiac Diseases

**DOI:** 10.3390/cells7120279

**Published:** 2018-12-19

**Authors:** Jin Li, Deli Zhang, Marit Wiersma, Bianca J. J. M. Brundel

**Affiliations:** Department of Physiology, Amsterdam UMC, Vrije Universiteit Amsterdam, Amsterdam Cardiovascular Sciences, 1081 HV Amsterdam, The Netherlands; d.zhang@vumc.nl (D.Z.); m.wiersma1@vumc.nl (M.W.)

**Keywords:** proteostasis, autophagy, cardiac disease, atrial fibrillation

## Abstract

Due to ageing of the population, the incidence of cardiovascular diseases will increase in the coming years, constituting a substantial burden on health care systems. In particular, atrial fibrillation (AF) is approaching epidemic proportions. It has been identified that the derailment of proteostasis, which is characterized by the loss of homeostasis in protein biosynthesis, folding, trafficking, and clearance by protein degradation systems such as autophagy, underlies the development of common cardiac diseases. Among various safeguards within the proteostasis system, autophagy is a vital cellular process that modulates clearance of misfolded and proteotoxic proteins from cardiomyocytes. On the other hand, excessive autophagy may result in derailment of proteostasis and therefore cardiac dysfunction. Here, we review the interplay between autophagy and proteostasis in the healthy heart, discuss the imbalance between autophagy and proteostasis during cardiac diseases, including AF, and finally explore new druggable targets which may limit cardiac disease initiation and progression.

## 1. The Proteostasis Network in the Heart

A healthy proteostasis network is essential, safeguarding the proper function of the heart by maintaining normal cellular metabolic function through adequate protein biosynthesis, folding, trafficking, and clearance [1,2,3]. Research evidence shows that healthy proteostasis is controlled by the protein quality control (PQC) system, which consists of chaperones, protein degradation systems such as the ubiquitin-proteosomal system (UPS) and autophagy, and the cytoskeleton. In the case of stress, multiple stress pathways are activated, including the heat shock response (HSR), the unfolded protein response (UPR), and oxidative stress response [3,4,5,6,7]. In response to proteotoxic stress in the cytosol, the HSR is activated. The HSR acts as the cell’s frontline system to safeguard the folding and trafficking of functional proteins by means of chaperones, including heat shock proteins (HSPs) and chaperonins (such as T-complex protein 1 ring complex [TRiC], HSPB1, HSPB5, HSPB6, HSPB7, and HSPB8), which are highly expressed in the heart [8,9,10]. Proteotoxic stress in the endoplasmic reticulum (ER) may lead to activation of the UPR, which prevents protein aggregation and guarantees correct protein folding. If the UPR fails and an accumulation of unfolded proteins occurs, the UPS will be activated. UPS-mediated proteolysis, a primary degradation system, can remove misfolded, oxidized, mutant, and damaged proteins from the cell [11,12]. After longer periods of cellular proteotoxic stress, the HSR, UPR, and UPS can become overwhelmed. In this case, autophagy is activated to clear misfolded proteins and thereby guarantee cell survival [2]. For the proper function of both autophagy and the pathways of the PQC system, an intact cardiomyocyte cytoskeleton is of crucial importance for communication between the components in the proteostasis network. In cardiomyocytes, the cytoskeleton is highly specialized and includes actin filaments, desmin (intermediate) filaments, and microtubules.

To summarize, a functional PQC system underlies healthy proteostasis in cardiomyocytes, and safeguards the maintenance of normal contractile function in the heart. Within the PQC system, autophagy is important for the clearance of stress-induced misfolded proteins, thereby guaranteeing cardiomyocyte survival.

## 2. Key Role for Autophagy in Proteostasis

### 2.1. Autophagy

Autophagy, a lysosome-mediated degradation pathway, plays a critical role in proteostasis by removing potentially toxic cytosolic protein aggregates and damaged organelles inside cells [13]. According to the mode of cargo delivery to the lysosome, autophagy has been categorized into three different types: macroautophagy, microautophagy, and chaperone-mediated autophagy (Figure 1) [14]. In macroautophagy, the formation of a double membrane structure, the autophagosome, is the first step. The autophagosome travels along microtubules and engulfs damaged organelles and aberrant proteins, followed by fusion with a lysosome to form the autolysosome. In autolysosomes, lysosomal hydrolases degrade the cargo into recyclable ATP, amino acids, and fatty acids (Figure 1a) [15,16]. This process is mediated by autophagy-related (ATG) proteins, which modulate the biogenesis of the autophagosome and its subsequent fusion with the lysosome. Recently, evidence is emerging showing that macroautophagy is a highly selective quality control mechanism, and its basal levels are important to maintain intracellular proteostasis. A number of organelles have been found to be turned over by macroautophagy in an organelle-selective manner, for example, targeting of mitochondria (mitophagy), endoplasmic reticulum (reticulophagy or ERphagy), peroxisomes (pexophagy), lipid droplets (lipophagy), and even portions of the nucleus (nucleophagy) [17]. In contrast to macroautophagy, microautophagy involves non-selective engulfment of cytoplasmic contents into the lysosomes (Figure 1b). Chaperone-mediated autophagy is unique in mammalian cells and requires two essential components, the cargo recognition complex in the cytosol, and the cargo translation complex at the lysosome. Hsc70, an important component of the cargo recognition complex, recognizes and attaches to a specific motif sequence in target proteins, whereafter the complex shuttles and binds to the lysosomal-associated membrane protein type 2A (LAMP2A). LAMP2A forms a lysosomal channel to translocate the target protein, with the help of a lysosomal-resident form of Hsc70 (lys-hsc70), to the lysosomal lumen for degradation (Figure 1c) [18,19]. In addition, mitophagy is an organelle-specific form of macroautophagy, which selectively removes damaged and dysfunctional mitochondria (Figure 1d) [20]. Since cardiomyocytes contain a very high volume of mitochondria, and cardiomyocytes are metabolically active cells, it is essential to conserve mitochondrial function to maintain a healthy energy balance and cardiac contractile function. This review focuses on macroautophagy (hereafter ‘autophagy’) and mitophagy, as they are the best studied in the heart and play a crucial role in the maintenance of cardiomyocyte homeostasis.

### 2.2. Key Regulators and Signaling Pathways of Autophagy in the Heart

Research has revealed that, dependent on the underlying stress condition, cardiomyocyte autophagy requires several key regulators and signaling pathways. Three master transcriptional regulators of autophagy have been recognized in the heart, including transcription factor EB (TFEB), zinc-finger protein with KRAB and SCAN domains 3 (ZKSCAN3), and bromodomain containing 4 (BRD4). TFEB, a master regulator of lysosomal pathways, regulates a wide range of autophagy-related genes [21]. TFEB overexpression is sufficient to induce autophagy. However, after autophagy activation in response to different stimuli, such as nutrient depletion (starvation), TFEB is dephosphorylated through the inhibition of mammalian target of rapamycin (mTOR), resulting in its rapid translocation to the nucleus and the activation of autophagy genes [22]. TFEB activation protects against cardiac proteotoxicity by increasing autophagic flux [23]. ZKSCAN3 represents the transcriptional counterpart of TFEB, since it represses the transcription of a number of autophagy-related genes, including Unc-51-like autophagy activating kinase 1 (Ulk1). Upon autophagy induction, ZKSCAN3 translocates from the nucleus to the cytoplasm, allowing the transcriptional activation of target genes by TFEB. Interestingly, ZKSCAN3 knockdown is sufficient to induce autophagy, whereas its overexpression can inhibit autophagy [24]. BRD4, an evolutionarily conserved autophagy repressor, inhibits autophagic and lysosomal activities by repressing the expression of autophagy and lysosome genes under nutrient-rich conditions, and de-represses this program to contribute to autophagy activation during nutrient deprivation [25]. Among numerous ATG proteins and kinases, mTOR is considered to be a critical negative regulator of autophagy. The activity of mTOR is regulated by the availability of nutrients, energy, and oxygen [26]. In nutrient-rich conditions, mTOR is activated, and thereby inhibits autophagy through phosphorylation, and subsequently, inactivation, of Ulk1/2, which inhibits autophagosome formation (Figure 2a) [27]. In addition, autophagy is also regulated through mTOR-independent mechanisms, including AMP-activated protein kinase (AMPK), hypoxia-inducible factor 1α (HIF1α), sirtuin 1 (sirt1), and ER stress (Figure 2b–d). As an energy sensor, AMPK is activated during metabolic stress that either interferes with ATP production (e.g., during glucose deprivation, hypoxia, and ischemia) or accelerates ATP consumption (e.g., during muscle contraction). Activation of AMPK requires the phosphorylation of its α subunit at Thr172 by two upstream kinases, liver kinase B1 (LKB1) and Ca^2+^/calmodulin-dependent kinase kinase (CaMKK)-β. Activation of AMPK initiates autophagy, not only through the inhibition of mTOR, but also by direct phosphorylation and, thereby, activation of Ulk1 (Figure 2b) [28]. Chronic AMPK activation restores cardiac function by upregulating autophagy activity in diabetic OVE26 mice (type I diabetes) [29]. HIF1α activates autophagy under hypoxia, where it provides a protective response, leading to cellular adaptation and survival (Figure 2c) [30]. Moreover, sirt1, an NAD^+^-dependent deacetylase, directly enhances autophagy in cells by deacetylating ATG proteins, resulting in expression of ATG proteins and autophagosome formation (Figure 2d) [31]. In fasting hearts, sirt1 indirectly stimulates autophagy by deacetylating transcription factors regulating autophagy, including Foxo family members, thereby maintaining left ventricular function during starvation [32]. In addition, NAD^+^ itself also influences autophagy (Figure 2e), as experimental evidence has revealed that autophagic flux is negatively affected by decreases in NAD^+^ levels during ischemia, due to the inhibition of lysosomal function [33]. Furthermore, nicotinamide phosphoribosyl transferase (Nampt), a key enzyme in the salvage pathway of NAD^+^ synthesis in cardiomyocytes, is downregulated in the heart in response to prolonged ischemia, which contributes to a decrease in the level of NAD^+^ in the heart, inhibition of autophagic flux, and, consequently, increase in cardiomyocyte death [34]. ER stress is also known to induce autophagy, in which dissociation of BiP/GRP78 leads to phosphorylation of PERK and eIF2α. The latter induces transcription of ATG12, an important protein in autophagosome formation (Figure 2f) [35]. Vacuolar protein sorting 34 (Vps34) is also a critical regulator of autophagy in the heart, as deletion of Vps34 can impair autophagy, which leads to accumulation of aberrant complexes of αB-crystallin and desmin in cardiomyocytes (Figure 2g) [36]. A rise in free cytosolic calcium ([Ca^2+^]c) is also a potent inducer of autophagy, and promotes autophagy by modulating downstream targets, such as inositol 1,4,5-trisphosphate receptor (IP3R), and calmodulin-dependent protein kinase II (CAMKII) (Figure 2h) [37]. The IP3R regulates autophagy through its interaction with Beclin1, and its inhibition strongly induces autophagy [38]. CaMKII contributes to autophagy by promoting the calcium-dependent activation of AMPK [39]. Activation of mitophagy is regulated by different regulators compared to autophagy. Mitophagy is mainly regulated by three proteins: PTEN-induced putative kinase protein 1 (PINK1), mitofusin 2 (Mfn2), and Parkin [40,41,42]. In cases where mitochondria are depolarized and damaged, this contributes to PINK1 accumulation at the outer mitochondrial membrane. PINK1 accumulation phosphorylates Mfn2, which promotes Parkin to bind Mfn2 in a PINK1-dependent manner, thereby targeting damaged mitochondria for mitophagy [42]. Interestingly, AMPK activation also serves a critical role in mitochondria quality control, via modulation of mitophagy in the heart. In failing hearts, PINK1 phosphorylation by AMPKα2 was essential for activation of mitophagy to prevent the progression of heart failure (HF) [43].

### 2.3. Autophagy in Proteostasis

Since cardiomyocytes are post-mitotic cells, and therefore not constantly replaced by proliferation, a proper PQC system is highly important to safeguard cardiomyocyte function. Since aberrant proteins and organelles may impair cardiomyocyte function, the cell activates a specialized autophagic protein degradation pathway to clear these proteotoxic elements. In addition, autophagy generates recycled fatty acids and amino acids, and provides energy and nutrients for the cardiomyocytes [44]. In all cell types, including cardiomyocytes, autophagy serves proteostasis by coordinating with other components of the proteostasis network. It is largely accepted that the UPS and autophagy collaborate in defending against proteotoxic stress, and that p62/SQSTM1, a sensor of proteotoxic stress, acts as a mediator in the crosstalk between the UPS and autophagy [45,46,47]. P62/SQSTM1 is a multifunctional protein, containing a number of protein–protein interaction motifs that are involved in the regulation of cellular signaling, protein aggregation, and degradation. Also, autophagy inhibition may impair proteasome function, thereby severely increasing the levels of proteasome substrates in the cardiomyocyte [48]. In addition, evidence suggests that the impairment of a specific degradation pathway can result, as part of compensatory mechanism, in the activation of another one [49,50]. Recently, more attention has been paid to the exact relationship between autophagy and the microtubule network. It was found that microtubule dynamics, posttranslational modifications of alpha-tubulin and microtubule molecular motors (on the outer face of microtubules) are important regulators within the autophagy process. Microtubule dynamics and microtubule molecular motors are involved in autophagosome formation, orchestration of pre-autophagosomal structures and autophagosome movements, and immature and mature autophagosome localization [15,51,52].

To summarize, autophagy contributes to the maintenance of cellular proteostasis by regulating the PQC. On the one hand, autophagy is able to degrade potentially toxic molecules and organelles from the cardiomyocyte, thereby preventing the accumulation of misfolded or aberrant proteins. On the other hand, autophagy acts as a cellular recycling program, reclaiming amino acids, lipids, and other molecular building blocks liberated from substrates with the help of lysosomal acidic hydrolases.

## 3. Autophagy in Cardiac Ageing and Diseases

As mentioned above, cardiomyocytes largely depend on efficient clearance of aberrant and damaged proteins and organelles via autophagy to maintain healthy heart function [53]. Therefore, impairment of autophagy results in an imbalance in protein degradation and handling of misfolded or damaged proteins, which has been associated with ageing, and onset and progression of cardiac disease. However, several cardiac diseases are linked to excessive activation of autophagy, which may lead to the degradation of contractile proteins and autophagic cell death, which is detrimental for cardiomyocytes (Table 1) [54].

### 3.1. Cardiac Ageing

Ageing is an independent risk factor for cardiac disease development [55]. Age-related changes in the proteostasis network are observed in cardiomyocytes. During normal ageing, the structure and function of the heart is diminished, resulting in a decline in diastolic heart function during rest and systolic function during exercise, impairment of Ca^2+^ homeostasis, induction of reactive oxygen species (ROS), and structural remodeling, including cardiac hypertrophy and fibrosis. Autophagy is downregulated in the heart during the course of ageing [56]. Stimulation of autophagy in the aged hearts of mice decreases hypertrophy, reduces protein damage, restores Ca^2+^ homeostasis, attenuates hypertrophy, and improves contractile function [56]. Cardiac-specific deletion of GSK-3α, a crucial regulator in age-related pathologies in mice, accelerated the development of cardiac ageing, accompanied by suppression of autophagy [57,58]. In addition, a genome-wide association study of ageing identified a single nucleotide polymorphism (SNP) near the ATG4C gene as being associated with a higher risk of death, suggesting that autophagy may be intimately involved in the risk of heart disease in elderly patients [59]. This suggests that autophagy is required for normal cardiac function during ageing, and that impairment of autophagy in the ageing heart may contribute to cardiac disease development and progression. Moreover, impaired autophagy affects the UPS, as damaged or aged proteasomes are degraded by autophagy, contributing to senescence of the heart and cardiac disease onset [48]. In recent years, the role of mitophagy in ageing has attracted attention. Mitochondrial dysfunction is a hallmark of ageing [60]. An experimental study in *Caenorhabditis elegans* revealed that mitochondria accumulate with age, which was related to a decrease in clearance of damaged mitochondria via mitophagy [61]. In aged mice, the induction of mitophagy was found to improve overall mitochondrial function and prevent arterial wall stiffness [62]. 

### 3.2. Cardiomyopathy

#### 3.2.1. Inherited Cardiomyopathy

Inherited cardiomyopathies are characterized by mutations in genes encoding sarcomeric proteins, and are associated with high mortality and morbidity worldwide [63]. There are four main types of inherited cardiomyopathy: hypertrophic cardiomyopathy (HCM), dilated cardiomyopathy (DCM), restrictive cardiomyopathy, and arrhythmogenic right ventricular cardiomyopathy. Accumulating evidence indicates that alterations in autophagy are a vital factor in pathological development of these diseases. Defective autophagy was reported in HCM caused by Danon disease, Vici syndrome, or LEOPARD syndrome. These cardiomyopathies are related to a mutation in a gene encoding a protein involved in the autophagy–lysosomal pathway, leading to protein accumulation [64,65,66]. Significantly, incomplete mitophagic flux and mitochondrial dysfunction are also shown in both in vitro and models of Danon disease [67]. Furthermore, altered autophagy was also found in HCM directly caused by mutations in *MYBPC3*, the most frequently mutated gene in HCM [68]. Moreover, data shows that autophagy was impaired in a MYBPC3-targeted knock-in mouse model, and that activation of autophagy ameliorated the cardiac disease phenotype in this mouse model [68]. Recently, it was shown that Vps34, an important autophagy regulator, was decreased in the myocardium of HCM patients, and muscle-specific deletion of Vps34 resulted in an HCM-like phenotype and sudden death in a mouse model [36]. Furthermore, deletion of Vps34 impaired autophagy in cardiomyocytes, as indicated by αB-crystallin-positive aggregates in mice with HCM-like abnormalities [36]. Interestingly, autophagy flux was also impaired in DCM caused by a Pleckstrin homology domain-containing protein (*PLEKHM2*) mutation [69]. Autophagic vacuoles in cardiomyocytes are associated with an improved heart failure prognosis in patients with DCM, suggesting that autophagy may play a role in the prevention of myocardial degeneration [70]. Autophagic vacuolization in DCM has also been reported for mutations in novel α-actinin 2 (*ACTN2*), *MYBPC3* and nebulette (*NEBL*) [71,72,73]. This observation suggests that the accumulation of autophagic vacuoles implies cardiomyocyte stress. However, the interpretation of vacuole accumulation remains unclear, since it could reflect an increase in autophagic activity or an impairment of autophagosome–lysosome fusion [74]. So far, inherited cardiomyopathies are associated with impairment of autophagy, and activation of autophagy ameliorates cardiac dysfunction.

#### 3.2.2. Diabetic Cardiomyopathy

Diabetic cardiomyopathy is characterized by ventricular dysfunction that increases the risk of heart failure and mortality in diabetic patients, independent of vascular pathology [75]. There is a strong indication that autophagy is involved in the pathophysiology of diabetic cardiomyopathy, however, the exact role of autophagy in diabetic cardiomyopathy remains controversial [29,76,77]. It has been reported that autophagic adaptations in diabetic cardiomyopathy differ between type 1 and type 2 diabetes [76]. Moreover, autophagy in the heart is enhanced in type 1 diabetes, but is suppressed in type 2 diabetes. Cardiac damage in the streptozotocin (STZ)-induced and OVE26 type 1 diabetic heart is ameliorated in Beclin1- or ATG16-deficient mice [78]. In high fat diet (HFD)-induced diabetic cardiomyopathy, increased mTORC1 activity contributes to the development of diabetic cardiomyopathy, and mTORC1 inhibition prevents the development of HFD-induced diabetic cardiomyopathy by improving hepatic insulin sensitivity in obesity [79]. Conversely, two studies show that autophagy is suppressed in the hearts of STZ-induced diabetic mice and OVE26 type 1 diabetic model mice [29,78]. One study from Xie et al. reports that suppressed autophagy in hearts of STZ-treated mice and OVE26 type 1 diabetic mice improves cardiac function by reducing AMPK activity [29]. Another study, from Xu et al., shows that diminished autophagy in the same models is an adaptive response, one that limits cardiac dysfunction in the type 1 diabetic heart by increasing mTORC1 activity [78]. Therefore, further research is needed to determine whether autophagy is beneficial or detrimental in diabetes. Interestingly, exercise has beneficial effects on human health, including protection against metabolic disorders such as obesity and diabetes [80]. BCL2 is a crucial regulator of exercise-induced autophagy in vivo, and autophagy induction may contribute to the improved metabolic effects of exercise, indicating that exercise has potential beneficial effects on diabetic cardiomyopathy [81]. 

### 3.3. Ischemic Heart Disease

#### 3.3.1. Atherosclerosis

Atherosclerosis is a progressive and complex disease that causes the buildup of plaque inside the walls of arteries. Due to technical limitations, investigation on the role of autophagy in plaque formation has not yet reached clear answers. Based on existing evidence, triglycerides or other dietary lipids are degraded via autophagy to provide free fatty acid substrates under physiological conditions [82]. However, in the heart, our understanding of the lipid-induced regulation of autophagy is still emerging. In vitro studies in H9C2 cardiomyocytes recapitulate some of the in vivo findings on lipid overload-induced cardiac autophagy [83]. Other studies showed that, in human endothelial cells, oxidized LDLs (OxLDLs) trigger the activation of autophagy by upregulating Beclin1 expression, which contributes to phagocytosis of OxLDLs exposed cells [84]. In an ApoE^−/−^ mice study, the importance of autophagy in atherosclerosis progression was further demonstrated by utilizing specific smooth muscle cells (SMCs) containing a deletion in ATG7. These SMCs exhibited accelerated atherosclerotic plaque development after 10 weeks of high fat diet [85]. Interestingly, it has been recently reported that PINK1 or Parkin knockdown increases the cytotoxic response of human vascular SMCs exposed to OxLDL, whereas PINK1 or Parkin overexpression has cytoprotective effects, suggesting that mitophagy plays a critical role in modulating vascular SMCs fate, whether by favoring cell survival or by enhancing apoptosis [86]. Furthermore, in a macrophage-specific ATG5 knockout mice model, the study shows that autophagy becomes dysfunctional in atherosclerosis, and its deficiency promotes atherosclerosis in part through inflammasome hyperactivation [87]. However, the relevance of beneficial effects of autophagy in the early stages of atherosclerosis, and the detrimental effects of autophagy observed in the late stages of mouse atherosclerotic plaque formation, remains to be further demonstrated in human clinical samples.

#### 3.3.2. Myocardial Infarction

Myocardial infarction (MI), one of the major contributors of morbidity and mortality in patients with coronary heart diseases (CHD) worldwide, is the irreversible death of cardiac muscle following a prolonged lack of oxygen supply (ischemia) [88]. In experimental cell models of hypoxia or starvation, the limitation in the availability of oxygen or nutrients in cardiomyocytes is a powerful activator of autophagy via HIF1-α and AMPK, respectively [30,89]. During an ischemic event in MI, limitations in the availability of both oxygen and nutrients are found to activate autophagy [90,91]. Inhibition of autophagy during ischemia leads to activation of the cell death program, suggesting that autophagy plays a cardio-protective role during ischemia. However, heart reperfusion after ischemia restores oxygen and nutrients to the injured tissue, but triggers a complex cascade of events and a second wave of injury [92]. The role for autophagy during reperfusion remains to be clarified, since it has not been thoroughly established whether autophagic flux is also increased during reperfusion [93]. Inhibition of autophagy significantly decreases infarct size and improves cardiac functions after ischemia/reperfusion (I/R)-induced myocardial injury in an I/R mouse model [58]. However, autophagic induction was detrimental for an infarcted heart during reperfusion, and its attenuation in Beclin1^+/−^ mice decreased in the infarcted area after I/R [90]. Since myocardial reperfusion is accompanied by an increase in ROS that triggers the upregulation of Beclin1 [94,95], which in turn has been associated with the onset of maladaptive autophagy in the heart [90,96], autophagy may be a double-edged sword in myocardial I/R injury. Interestingly, progressive reduction in cardiomyocyte autophagy in the remote non-infarcted myocardium was associated with myocardial oxidative stress and left ventricular remodeling after MI. Antioxidants prevented the reduction in cardiomyocyte autophagy after MI, suggesting that oxidative stress mediates reduction in cardiomyocyte autophagy that contributes to post-MI remodeling [97]. In addition, selective elimination of damaged mitochondria by mitophagy is predicted to protect cardiomyocytes during reperfusion. Consistent with this, loss of Pink1 has been reported to increase the infarct size after I/R [98]. Furthermore, Parkin^−/−^ mice had reduced survival and developed larger infarcts when compared with wild type mice after MI, and Parkin^−/−^ cardiomyocytes had reduced mitophagy and dysfunctional mitochondria after infarction. Overexpression of Parkin in isolated cardiomyocytes also protected against hypoxia-mediated cell death, suggesting that Parkin plays a critical role in adapting to stress in the myocardium by promoting removal of damaged mitochondria [99]. 

### 3.4. Atrial Fibrillation

Atrial fibrillation (AF) is the most common progressive cardiac rhythm disorder, and is associated with substantial morbidity and mortality. It has been recognized that AF persistence is rooted in the presence of proteostasis derailment in the cardiomyocyte [100]. Recently, studies revealed a crucial role of autophagy in proteostasis derailment contributing to AF progression. We observed that autophagy is induced upon endoplasmic reticulum (ER) stress, and is associated with cardiomyocyte remodeling in experimental and human AF. Inhibition of ER stress was shown to attenuate autophagy and to protect against cardiac remodeling in in vitro and in vivo models of AF [101]. In addition, a study reported that AMPK-dependent autophagy occurred in atrial cardiomyocytes after rapid atrial pacing of dogs and in persistent AF patients, indicating that activation of AMPK and downstream autophagy may also be a novel mechanistic contributor to AF [102]. Ca^2+^ deregulation is a critical hallmark of cardiac arrhythmias and dysfunction of the regulatory proteins involved in Ca^2+^ homeostasis, including ryanodine receptor type 2 (RyR2), IP3R, and CAMKII, which may lead to the development of AF. CAMKII activation and Ca^2+^ release from the ER can activate autophagic pathways, indicating that both of them can modulate autophagy activation in cardiomyocytes [37]. Induction of autophagy has been observed in atrial cardiomyocytes in AF patients with severe mitral and tricuspid regurgitation, and is closely associated with the degradation of sarcomeric structures (myolysis) in this disease [54]. Autophagic flux and ATG7 protein levels were markedly increased in atria of persistent AF patients and a rabbit model of rapid atrial pacing [103]. So far, studies addressing the role of mitophagy during AF are lacking. In addition, recent evidence also showed that impaired cardiac autophagy is present in patients developing postoperative atrial fibrillation (POAF) after coronary artery bypass surgery [104]. Taken together, the findings in experimental AF models and clinical AF indicate that excessive activation of autophagy may result in cardiomyocyte impairment and myolysis.

### 3.5. Heart Failure

Heart failure (HF) is a clinical syndrome caused by structural and functional defects in the myocardium, resulting in impairment of ventricular filling and reduction in the ejection fraction [105]. The first stage of HF exists as compensatory left ventricular hypertrophy, which develops to chronic HF. Autophagy has been recognized to play a role in the pathophysiology of HF. In the heart subjected to thoracic transverse aortic constriction (TAC), an experimental model of HF, autophagy is initially suppressed in hypertrophied hearts. During development of chronic HF, autophagy becomes activated through the upregulation of Beclin1 [96,106]. Initially, activation of autophagy prevents TAC-induced ventricular hypertrophy by increasing protein degradation, thereby improving ventricular function [107,108]. The study shows that the protein level of PINK1 is decreased after TAC. In line, PINK1-deficient mice develop age-dependent hypertrophy and cardiac dysfunction accompanied by mitochondrial dysfunction, even without pressure overload, and consequently suppression of mitophagy resulting in impaired heart function [109]. However, excessive autophagy induction in a failing heart leads to autophagic cell death and loss of cardiomyocytes in ischemic cardiomyopathy (ICM) or HF, suggesting that the level and duration of autophagy determines whether autophagy is protective or detrimental in HF [110,111,112,113]. 

To summarize, autophagy has distinct roles in various cardiac diseases. Whether it is beneficial or detrimental depends on the underlying pathological condition of cardiac disease. Excessive activation of autophagy, as observed in HF and AF, promotes degradation of contractile protein in the cardiomyocytes, contributing to cardiomyocyte damage and autophagic cell death. In case of cardiomyopathy, caused by expression of a mutant protein, autophagy leads to removal of misfolded proteins, thereby contributing to cardiomyocyte function.

## 4. Autophagy as Potentially Therapeutic Target in Cardiac Disease

As mentioned above, autophagy shows distinct roles in different pathological conditions in the heart. Therefore, in the following section we will discuss potential pharmacological modulators of autophagy for the treatment of specific cardiac pathological conditions (Table 2).

### 4.1. Therapeutic Modulation of Autophagy during Ageing

As mentioned before, autophagy is intimately involved in the regulation of lifespan and ageing [114]. Autophagy is downregulated in the heart with age, and thereby contributes to development of cardiac disease. Therefore, activation of autophagy may be used to delay ageing of the heart and thereby attenuate cardiac disease development. Lifestyle change by caloric restriction (CR) has been shown to extend lifespan and reduce age-related cardiac pathologies by stimulating autophagy [115,116,117]. Interestingly, accumulating evidence also shows that the major effect of CR on autophagy is the modulation of multiple upstream regulators of autophagy, including sirt1, AMPK, and mTOR. CR increases the expression and activity level of sirt1, and pharmacological activation of sirt1 by resveratrol mimics important outcomes of CR, including the reduction of age-related cardiac dysfunction [118,119]. The AMPK activator metformin has been shown to attenuate cardiomyocyte contractile defects in an ageing-induced myocardial contractile dysfunction model [120,121]. Direct suppression of mTOR via administration of rapamycin inhibits the adverse effects of ageing, increases lifespan, and promotes autophagy in the heart as well as in many other cell types and organs, even when autophagy was suppressed by ageing [122,123]. Supplementation with spermidine, a natural polyamine, has shown cardiac protective effects and lifespan extension by enhancing autophagy in ageing-related skeletal muscle atrophy in young and old mice [124,125].

### 4.2. Pharmacological Modulation of Autophagy in Inherited Cardiomyopathy

Studies show that autophagy is impaired in HCM and DCM. In mice with HCM caused by aggregation of the αB-crystallin mutant, ATG7-dependent activation of autophagy reduces accumulation of amyloid oligomers in cardiomyocytes [96]. Moreover, inhibition of autophagy through beclin1 knockout accelerates ventricular dysfunction in the αB-crystallin mutant mice, suggesting that stimulation of autophagy may improve cardiac function and reduce ventricular remodeling [126,127]. Rapamycin treatment and CR both activate autophagy and improve the HCM phenotype in a mouse model of HCM, indicating that it is of interest to test whether mTOR inhibitors and AMPK modulators protect against cardiac remodeling and ventricular dysfunction in the heart with early stage HCM or DCM by stimulating autophagy [68]. However, it should be noted that excessive activation of autophagy is not beneficial, as observed in end-stage HCM and DCM patients [128,129].

### 4.3. Pharmacological Modulation of Autophagy in MI

It has been recognized that cardiac autophagy is involved in the pathological process of MI. As we mentioned above, activation of autophagy may be protective in the heart during ischemia. Trehalose, a natural and non-reducing disaccharide, attenuates cardiomyocyte death by activating autophagy during glucose deprivation in a model mimicking ischemia [130]. Similarly, trehalose administration improves cardiac remodeling after MI through the activation of autophagy in the mouse heart [131]. These studies suggest that trehalose may also be considered as an alternative autophagy inducer for the treatment of cardiac ischemic injury. Interestingly, antihypertensive drugs, including propranolol, verapamil, nicardipine, and nimodipine, not only ameliorate ischemic tolerance and reduce blood pressure, but also stimulate cardiomyocyte autophagy, suggesting that administration of these drugs can provide additional cardioprotection during ischemia [74,132]. During reperfusion, urocortin, an endogenous cardiac peptide, inhibits autophagy by decreasing Beclin1 expression, indicating administration of urocortin as an interesting treatment option [133,134]. In addition, administration of chloroquine, a strong inhibitor of autophagic flux, is indicated to delay autophagy-induced degradation of proteins, such as catalases, that are essential for the myocardial response to reperfusion injury [135]. Propofol, a common drug used for induction of anesthesia, has both antioxidant and autophagy inhibiting properties, and limits myocardial damage during reperfusion injury. Notably, propofol inhibits autophagy through inhibition of Beclin1 and activation of mTOR [136]. Interestingly, two antimicrobial agents, chloramphenicol and sulfaphenazole, have recently been shown to activate autophagy and reduce myocardial damage during IR [137,138].

### 4.4. Pharmacological Modulation of Autophagy in AF

There are strong indications that alterations of autophagy contribute to AF pathogenesis. Therefore, pharmacological modulation of autophagy may offer a novel and promising strategy to prevent or treat AF. We found that blocking of ER stress-associated autophagy by the chemical chaperone 4-phenyl butyrate prevents downstream autophagy activation and contractile dysfunction in in vitro and in vivo models of AF, suggesting 4-phenyl butyrate as a potential drug for the treatment of AF [101]. Accordingly, Ca^2+^-channel blockers and antiarrhythmic drugs in the clinic, such as verapamil, nimodipine, nitrendipine, niguldipine, and pimozide, may modulate autophagy by decreasing intracellular Ca^2+^ levels [132]. In addition, ageing is an important risk factor for the development of AF. Therefore, the discussed therapies to attenuate cardiac ageing, such as CR, may have beneficial effects in the prevention of risk factors which underlie AF development.

### 4.5. Pharmacological Modulation of Autophagy in Heart Failure

Activation of autophagy during compensated hypertrophy in response to moderate pressure overload was found to reveal protective effects against further development of HF. Studies show that a low-dose of rapamycin administration or AMPK activators, including metformin and 5-aminoimidazole 1 carboxamide ribonucleoside (AICAR), improve cardiac function, reduce cardiac hypertrophy, and delay the onset of HF during pressure overload [108,139,140]. In addition, metformin, a glucose-lowering agent, is prescribed to patients with diabetes who display impaired cardiac activity of AMPK [141]. In contrast, autophagy suppression was found to be beneficial in the presence of severe pressure overload and in end-stage failing hearts. Urocortin, a protein that belongs to the family of corticotropin-releasing factors, was shown to inhibit Beclin1 and autophagy, and thus could be a valid therapeutic option [134]. In addition, excessive autophagy in HF may trigger cell death through massive depletion of essential proteins and organelles, suggesting that lysosomal enzyme inhibitors, such as chloroquine, could be appropriately used in this condition to delay autophagy-induced protein degradation and to reduce cell death [142]. 

To summarize, converse roles for pharmacological modulators of autophagy in the treatment of cardiac diseases in experimental models have been observed. Dependent on the mechanisms driving the pathological condition in various cardiac diseases, an inhibitor or an activator of autophagy is the treatment of choice. 

## 5. Conclusions

Maintaining proteostasis through autophagy is critical and essential in cardiac health. Autophagy is altered under different pathological conditions, resulting in an imbalance in protein degradation and/or impaired handling of misfolded/damaged proteins, which is often associated with cardiac disease. Autophagy is a so-called double-edged sword, where autophagic activation is a friend in some cardiac diseases, but a foe in others. Pharmacological modulation of autophagy offers a novel and promising strategy for treating cardiac diseases, including AF. However, there are several factors to consider when choosing the pharmacological modulator of autophagy. Firstly, autophagy has distinct roles in cardiac diseases, therefore, activators or inhibitors of autophagy should be chosen based on the mechanisms involved in the pathology of the cardiac disease. Secondly, excessive activation or complete inhibition of autophagy are both detrimental in cardiac disease development. Thirdly, it should be noted that, although several autophagy modulators are marketed, the efficacy of pharmacological modulation of autophagy in cardiac diseases is based exclusively on findings from experimental model systems. It is recommended that some exciting preclinical results be translated to human clinical trials.

## Figures and Tables

**Figure 1 cells-07-00279-f001:**
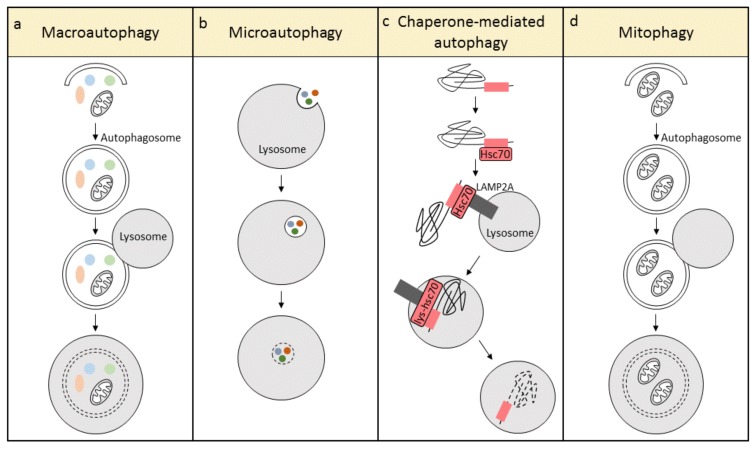
The three different types of autophagy: (**a**) macroautophagy, (**b**) microautophagy, and (**c**) chaperone-mediated autophagy. Mitophagy (**d**) is a specialized form of macroautophagy, which selectively degrades mitochondria.

**Figure 2 cells-07-00279-f002:**
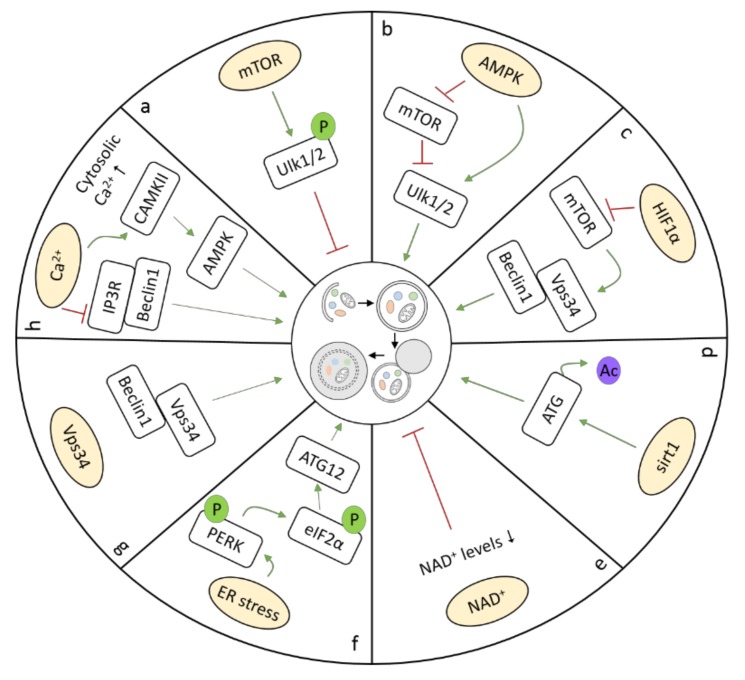
Different pathways activate autophagy: (**a**) mTOR, (**b**) AMPK, (**c**) HIF1α, (**d**) sirt1, (**e**) NAD^+^, (**f**) ER stress, (**g**) Vps34, and (**h**) Ca^2+^.

**Table 1 cells-07-00279-t001:** Autophagy in cardiac ageing and disease.

Condition	Subgroup	Main Pathways	Autophagy	Protective/Detrimental
Cardiac Ageing		Sirtuins, NAD^+^, AMPK and mTOR	↓	Detrimental
Inherited Cardiomyopathy	HCM	VSP34, ATG7	↓	Detrimental
	DCM		↓	Detrimental
Diabetic Cardiomyopathy	Type I	Beclin1, ATG16, AMPK, mTORC1	Not clear	Not clear
	Type II	mTORC1, BCL2	↓	Detrimental
Atherosclerosis		Beclin-1, ATG7, PINK1/Parkin	↓	Protective
MI	Ischemia	AMPK, mTORC1, NAD^+^, PINK1/Parkin	↓	Protective
	Reperfusion	Beclin1, mTOR, PINK1/Parkin	↓	Detrimental
AF	Persistent	ER, Ca^2+^, AMPK	↓	Detrimental
	POAF		↓	Not clear
Heart Failure	Hypertrophic stage	AMPK, mTOR, PINK1	↓	Detrimental
	Chronic	AMPK, Beclin1	↓	Detrimental

**Table 2 cells-07-00279-t002:** Pharmacological targeting of autophagy in cardiac ageing and disease.

Condition	Autophagy	Drug	Regulator
Cardiac Ageing	↓	Caloric restriction	Sirt1, AMPK, mTOR modulation
		Resveratrol	Sirt1 activation
		Metformin	AMPK activation
		Rapamycin	mTOR suppression
		Spermidine	Autophagy activation
Hypertrophic Cardiomyopathy	↓	Rapamycin	mTOR suppression
		Caloric restriction	Sirt1, AMPK, mTOR modulation
Myocardial Infarction-Ischemia	↑	Trehalose	Autophagy activation
		Antihypertensive drugs	
		Cloramphenicol, Sulfaphenazole	
Myocardial Infarction-Reperfusion	↑	Urocortin	Beclin1 suppression
		Cloroquine	Lysosomal enzyme suppression
		Propofol	Beclin1 suppression, mTOR activation
Atrial Fibrillation-Persistent	↑	Ca^2+^ channel blockers	Decrease intracellular Ca^2+^ levels
		Antiarrhythmic drugs	
		4-phenyl butyrate	ER stress suppression
Heart Failure-Hypertrophic Stage	↓	Rapamycin	mTOR suppression
		Metformin	AMPK activation
		AICAR	
Heart Failure-Chronic	↑	Urocortin	Beclin1 suppression
		Cloroquine	Lysosomal enzyme suppression

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
