# Peer review of "Role of Autophagy in Proteostasis: Friend and Foe in Cardiac Diseases"

_cells, 2018, doi:10.3390/cells7120279_

Round 1
Reviewer 1 Report
The autophagic system of proteolysis regulates numerous functions involved in cardiac physiology and pathology. The identification of autophagic gene mutations resulting in loss of function of several autophagy signalling components leads to the pathology of numerous CVD. A systematic understanding of the mechanisms of autophagic impairment in different models of metabolic inflexibility, including cellular and animal models as well as human failing ventricular samples should fuel discovery of novel cardiac-specific therapeutics targeting molecular components of autophagy system.
Specific changes:
Limited information on how different organelles participating in autophagy fail during cardio-metabolic dysfunction and how this process impact overall process of autophagy and initiation/progression of heart disease.
Information and reference to metabolic pathways which alter autophagy and trigger organelle specific damage is lacking.
Genetic diseases with CV phenotype in which autophagy is impaired is not discussed in this review such as Dannon disease.
Lacking discussion on transcriptional regulators of autophagy.
2015-2018 papers from mainstream authors who engage in researching cardiac autophagy are not cited in this review such as the following labs; Abel ED, Diwan A, Pulinilkunnil T. This needs to be addressed as these studies are pertinent to this review.
General changes:
1. Integration of the physiology/cell biology with experimental findings is essential. In other words for a critical piece of data, support the statement with a mouse model and highlight its phenotype.
2. A section highlighting autophagy initiation overview in different organelles will be helpful to the readers (e.g. mitochondria and mitophagy, peroxisome and pexophagy, nucleus and nucleophagy in the realm of cardiomyocyte or heart)
3. A section on AMPK and autophagy is desirable, information is missing on mouse models of AMPK modification and autophagy, and upstream kinases of AMPK and its impact on autophagy. Discussion on these points will add more “meat to the gravy”.
4. While elaborating on cardiomyocyte stressors and autophagy, the addition of calcium and autophagy, lipids and autophagy, exercise and autophagy would strengthen the review
5. The value of this review could be augmented by providing a table showing autophagy gene mutations in the heart, accumulating polypeptide, phenotype and animal model associated with that mutation and if possible human equivalent.
6. Authors can elaborate on mitophagy in the heart during various CV diseases. Metabolic control of mitophagy is poorly described. Given the role of energy metabolism in regulating cellular ATP, it is logical that the metabolic control of mitophagy is pertinent to this review. Additional details of which need to be described in this review. E.g. the role of glucose and fatty acid metabolism in regulating mitophagy? How amino acid regulation of mitophagy takes place, specifically during starvation (ATP loss) and in diseases such as ischemia and hypertrophy.
7. Transcriptional regulation of autophagy and lysosome function is not adequately discussed in this review. E.g. role of TFEB, ZKSCAN, BRD4 in regulating mitophagy, lysosomal function and autophagy and providing recent references supporting this work.
8. The figures presented in this review are indeed valuable. This review could be supported by a table showing autophagy gene mutations in the heart, accumulating polypeptide, phenotype and animal model associated with that mutation and if possible human equivalent.
9. While elaborating on cardiomyocyte stressors and mitophagy, discussion on the influence of calcium, lipids and exercise would strengthen the review
10. In the section on AMPK and mitophagy, information is missing on mouse models of AMPK modification and mitophagy, and upstream kinases of AMPK and its impact on autophagy.
11. The minor point is that the figures are not called out much and are not used effectively to guide the reader. Given the complexity of the text and processes being described, labelling up the figures into subsections and calling them out frequently to follow the text would significantly help the reader through this data. Please label the pathways in the figures in alphabetical order and refer to them in the text.
Author Response
Reviewer 1
The autophagic system of proteolysis regulates numerous functions involved in cardiac physiology and pathology. The identification of autophagic gene mutations resulting in loss of function of several autophagy signaling components leads to the pathology of numerous CVD. A systematic understanding of the mechanisms of autophagic impairment in different models of metabolic inflexibility, including cellular and animal models as well as human failing ventricular samples should fuel discovery of novel cardiac-specific therapeutics targeting molecular components of autophagy system.
Specific changes:
Limited information on how different organelles participating in autophagy fail during cardio-metabolic dysfunction and how this process impacts overall process of autophagy and initiation/progression of heart disease.
Thank you for this comment. Although we agree that information on interactions between different organelles participating in autophagy, especially in cardiac diseases, is of interest, the focus of our current review manuscript is directed at the interplay between autophagy and proteostasis. The main theme is to describe how altered autophagy affects protestasis (in terms of protein misfolding, and clearance) and whether autophagy acts as a friend or foe in various cardiac diseases. A detailed description how different organelles participate in autophagy and especially in cardiometabolic dysfunction is beyond the scope of the current manuscript.
We mentioned the interaction of different organelles in autophagy at page 2 lines 59-64.
‘Recently, evidence is emerging showing that macroautophagy is a highly selective quality control mechanism and its basal levels are important to maintain intracellular proteostasis. A number of organelles have been found to be turned over by marcoautophagy in an organelle-selective manner, for example, targeting of mitochondria (mitophagy), endoplasmic reticulum (reticulophagy or ERphagy), peroxisomes (pexophagy), lipid droplets (lipophagy), and even portions of the nucleus (nucleophagy) [17].’
Information and reference to metabolic pathways which alter autophagy and trigger organelle specific damage is lacking.
We added additional information on metabolic pathways and references to the manuscript.
We added information on page 4, lines 110-117.
‘As an energy sensor, AMPK is activated during metabolic stress, that either interferes with ATP production (e.g., during glucose deprivation, hypoxia, and ischemia) or accelerates ATP consumption (e.g., during muscle contraction). Activation of AMPK requires the phosphorylation of its α subunit at Thr172 by two upstream kinases, liver kinase B1 (LKB1) and Ca2+/calmodulin-dependent kinase kinase (CaMKK)-β. Activation of AMPK initiates autophagy, not only through the inhibition of mTOR, but also by direct phosphorylation and, thereby, activation of Ulk1 (Figure 2b) [28]. Chronic AMPK activation restores cardiac function by upregulating autophagy activity in diabetic OVE26 mice (type I diabetes) [29].
Page 4, Lines 146-149:
‘Interestingly, AMPK activation also serves a critical role in mitochondria quality control via modulation of mitophagy in the heart. In failing hearts, PINK1 phosphorylation by AMPKα2 was essential for activation of mitophagy to prevent the progression of heart failure (HF) [43].’
Page 7, Lines 208-212:
‘In recent years, the role of mitophagy in ageing has attracted attention. Mitochondrial dysfunction is a hallmark of ageing [60]. An experimental study in C. elegans revealed that mitochondria accumulate with age, which was related to a decrease in clearance of damaged mitochondria via mitophagy [61]. In aged mice, the induction of mitophagy was found to improve overall mitochondrial function and prevent arterial wall stiffness [62].’
Genetic diseases with CV phenotype in which autophagy is impaired is not discussed in this review such as Dannon disease.
Thank you for this comment. We added information on several genetic diseases with a cardiac disease phenotype such as Danon disease, Vici syndrome and LEOPARD syndrome to section 3.2.1. The studies show that autophagy is impaired in these diseases.
Page 7, Lines 220-225:
‘Defective autophagy was reported in HCM caused by Danon disease, Vici syndrome and LEOPARD syndrome. These cardiomyopathies are related to a mutation in a gene encoding a protein involved in the autophagy-lysosomal pathway, leading to protein accumulation [64-66]. Interestingly, incomplete mitophagic flux and mitochondrial dysfunction were observed in both in vitro and in vivo models of Danon disease [67]. Furthermore, altered autophagy was also found in HCM directly caused by mutations in MYBPC3, the most frequently mutated gene in HCM [68].’
Lacking discussion on transcriptional regulators of autophagy.
We improved elaboration on transcriptional regulators of autophagy in section 2.2. Here, several master transcriptional regulators of autophagy in the heart, including TFEB, mTORC1, ZKSCAN3 and BRD4 are discussed.
Page 3-4 Lines 86-103:
‘Research revealed that, dependent of the underlying stress condition, cardiomyocytes respond to stress by activating specific key regulators and signaling pathways of autophagy. Three master transcriptional regulators of autophagy have been recognized in the heart, including transcription factor EB (TFEB), zinc-finger protein with KRAB and SCAN domains 3 (ZKSCAN3) and bromodomain containing 4 (BRD4). TFEB, a master regulator of lysosomal pathways, regulates a wide range of autophagy-related genes [21]. TFEB overexpression is sufficient to induce autophagy. However, after autophagy activation in response to different stimuli, such as nutrient depletion (starvation), TFEB is dephosphorylated through the inhibition of mTOR, resulting in its rapid translocation to the nucleus and activation of autophagy genes [22]. TFEB activation protects against cardiac proteotoxicity via increasing autophagic flux [23]. ZKSCAN3 represents the transcriptional counterpart of TFEB, since it represses the transcription of a number of autophagy-related genes, including Unc-51-like autophagy activating kinase 1 (ULK1). Upon autophagy induction, ZKSCAN3 translocates from the nucleus to the cytoplasm, allowing the transcriptional activation of target genes by TFEB. Interestingly, ZKSCAN3 knockdown is sufficient to induce autophagy, whereas its overexpression can inhibit autophagy [24]. BRD4, an evolutionarily conserved autophagy repressor, inhibits autophagic and lysosomal activities by repressing the expression of autophagy and lysosome genes under nutrient-rich conditions, and derepression of this program contributes to autophagy activation during nutrient deprivation [25].’
2015-2018 papers from mainstream authors who engage in researching cardiac autophagy are not cited in this review such as the following labs; Abel ED, Diwan A, Pulinilkunnil T. This needs to be addressed as these studies are pertinent to this review.
Thank you for your suggestion. We cited the paper from Abel ED, because this paper provides information on the relation between lipids and autophagy.
General changes:
1. Integration of the physiology/cell biology with experimental findings is essential. In other words for a critical piece of data, support the statement with a mouse model and highlight its phenotype.
We added at several occasions in the manuscript information on model systems used to base study outcomes.
2. A section highlighting autophagy initiation overview in different organelles will be helpful to the readers (e.g. mitochondria and mitophagy, peroxisome and pexophagy, nucleus and nucleophagy in the realm of cardiomyocyte or heart)
We added information on autophagy of different organelles to page 2, Lines 59-64:
‘Recently, evidence is emerging showing that macroautophagy is a highly selective quality control mechanism and its basal levels are important to maintain intracellular proteostasis. A number of organelles have been found to be turned over by marcoautophagy in an organelle-selective manner, for example, targeting of mitochondria (mitophagy), endoplasmic reticulum (reticulophagy or ERphagy), peroxisomes (pexophagy), lipid droplets (lipophagy), and even portions of the nucleus (nucleophagy) [17].’
3. A section on AMPK and autophagy is desirable, information is missing on mouse models of AMPK modification and autophagy, and upstream kinases of AMPK and its impact on autophagy. Discussion on these points will add more “meat to the gravy”.
Thanks for your comments. We added information on how AMPK regulates autophay in mouse models and how upstream kinases of AMPK, such as LKB1 and CaMKK-β, affect autophagy to section 2.2.
See page 4, lines 110-117.
‘As an energy sensor, AMPK is activated during metabolic stress, that either interfere with ATP production (e.g., during glucose deprivation, hypoxia, and ischemia) or accelerate ATP consumption (e.g., during muscle contraction). Activation of AMPK requires the phosphorylation of its α subunit at Thr172 by two upstream kinases, liver kinase B1 (LKB1) and Ca2+/calmodulin-dependent kinase kinase (CaMKK)-β. Activation of AMPK initiates autophagy, not only through the inhibition of mTOR, but also by direct phosphorylation and, thereby, activation of Ulk1 (Figure 2b) [28]. Chronic AMPK activation restores cardiac function by upregulating autophagy activity in diabetic OVE26 mice (type I diabetes) [29].
4. While elaborating on cardiomyocyte stressors and autophagy, the addition of calcium and autophagy, lipids and autophagy, exercise and autophagy would strengthen the review
We added information how lipids regulate autophagy in section 3.3.1, and how exercise regulates autophagy in section 3.2.2. We also elaborate on calcium regulation of autophagy in section 3.4.
Page 8-9 Lines 268-289:
‘3.3.1 Atherosclerosis
Atherosclerosis is a progressive and complex disease that causes the buildup of plaques inside the wall of arteries. Due to technical limitations, investigation on the role of autophagy in plaque formation is not solved yet. Based on existing evidence, triglycerides or other dietary lipids are degraded via autophagy to provide free fatty acid substrates under physiological conditions [82]. However, in the heart, our understanding of the lipid-induced regulation of autophagy is still emerging. In vitro studies in H9C2 cardiomyocytes recapitulate some of the in vivo findings on lipid overload-induced cardiac autophagy [83]. Other studies showed that in human endothelial cells, oxidized LDLs (OxLDLs) trigger the activation of autophagy by upregulating Beclin1 expression, which contributes to phagocytosis of oxLDLs exposed cells [84]. In an ApoE−/− mice study, the importance of autophagy in atherosclerosis progression was further demonstrated by utilizing specific smooth muscle cells (SMC) containing a deletion in Atg7. These SMCs exhibited accelerated atherosclerotic plaque development after 10 weeks of high fat diet [85]. Interestingly, it has been recently reported that PINK1 or Parkin knockdown increases the cytotoxic response of human vascular SMC exposed to OxLDL, whereas PINK1 or Parkin overexpression has cytoprotective effects, suggesting that mitophagy plays a critical role in modulating vascular SMC fate by favoring cell survival or by enhancing apoptosis [86]. Furthermore, in a macrophage-specific ATG5 knockout mice model, the study shows that autophagy becomes dysfunctional in atherosclerosis and its deficiency promotes atherosclerosis in part through inflammasome hyperactivation [87]. However, the relevance of beneficial effects of autophagy in the early stages of atherosclerosis and the detrimental effects of autophagy observed in the late stages of mouse atherosclerotic plaque formation, remains to be further demonstrated in human clinical samples.’
Page 8, Lines 262-266:
‘Interestingly, exercise has beneficial effects on human health, including protection against metabolic disorders such as obesity and diabetes [80]. BCL2 is a crucial regulator of exercise-induced autophagy in vivo, and autophagy induction may contribute to the improved metabolic effects of exercise, indicating that exercise has potential beneficial effects in diabetic cardiomyopathy [81].’
Page 10, Lines 331-336:
‘Ca2+ deregulation is a critical hallmark of cardiac arrhythmias and dysfunction of the regulatory proteins involved in Ca2+ homeostasis, including ryanodine receptor type 2 (RyR2), IP3R and calmodulin-dependent protein kinase II (CAMKII), may lead to the development of AF. CAMKII activation and Ca2+ release from the ER can activate autophagic pathways, indicating that both of them can modulate autophagy activation in cardiomyocytes [37].’
5. The value of this review could be augmented by providing a table showing autophagy gene mutations in the heart, accumulating polypeptide, phenotype and animal model associated with that mutation and if possible human equivalent.
Thank you for this comment. So far, a limited amount of research findings on the role of autophagy mutations and development of heart diseases is available. We provide information on autophagy gene mutations and diseases such as Danon disease, Vici syndrome and LEOPARD syndrome in the text at page 7, Lines 220-225:
‘Defective autophagy was reported in HCM caused by Danon disease, Vici syndrome and LEOPARD syndrome. These cardiomyopathies are related to a defect in a gene encoding a protein involved in the autophagy-lysosomal pathway, leading to protein accumulation [64-66]. Interestingly, incomplete mitophagic flux and mitochondrial dysfunction were observed in both in vitro and in vivo models of Danon disease [67]. Furthermore, altered autophagy was also found in HCM directly caused by the mutations in MYBPC3, the most frequently mutated gene in HCM [68].’
6. Authors can elaborate on mitophagy in the heart during various CV diseases. Metabolic control of mitophagy is poorly described. Given the role of energy metabolism in regulating cellular ATP, it is logical that the metabolic control of mitophagy is pertinent to this review. Additional details of which need to be described in this review. E.g. the role of glucose and fatty acid metabolism in regulating mitophagy? How amino acid regulation of mitophagy takes place, specifically during starvation (ATP loss) and in diseases such as ischemia and hypertrophy.
We elaborated on the role of mitophagy during various cardiovascular diseases. However, studies describing the role of amino acids in the regulation of mitophagy in heart diseases, such as ischemia and hypertrophy, are scarce.
Page 10, Lines 338-342:
‘Autophagic flux and ATG7 protein levels were markedly increased in atria of persistent AF patients and a rabbit model of rapid atrial pacing [103]. So far, studies addressing the role of mitophagy during AF are lacking. In addition, recent evidence also showed that impaired cardiac autophagy is present in patients developing postoperative atrial fibrillation (POAF) after coronary artery bypass surgery [104].’
Page 10, Lines 354-358:
‘The study shows that the protein level of PINK1 is decreased after TAC. In line, PINK1-deficient mice develop age-dependent hypertrophy and cardiac dysfunction accompanied by mitochondrial dysfunction, even without pressure overload, and consequently suppression of mitophagy resulting in impaired heart function [109].’
7. Transcriptional regulation of autophagy and lysosome function is not adequately discussed in this review. E.g. role of TFEB, ZKSCAN, BRD4 in regulating mitophagy, lysosomal function and autophagy and providing recent references supporting this work.
Thank you for this comment. We added information on several master transcriptional regulators of autophagy to section 2.2, and also provided recent references to support the statements in this review.
Page 3-4 Lines 86-103:
‘Research revealed that, dependent of the underlying stress condition, cardiomyocytes respond to stress by activating specific key regulators and signaling pathways of autophagy. Three master transcriptional regulators of autophagy have been recognized in the heart, including transcription factor EB (TFEB), zinc-finger protein with KRAB and SCAN domains 3 (ZKSCAN3) and bromodomain containing 4 (BRD4). TFEB, a master regulator of lysosomal pathways, regulates a wide range of autophagy-related genes [21]. TFEB overexpression is sufficient to induce autophagy. However, after autophagy activation in response to different stimuli, such as nutrient depletion (starvation), TFEB is dephosphorylated through the inhibition of mTOR, resulting in its rapid translocation to the nucleus and activation of autophagy genes [22]. TFEB activation protects against cardiac proteotoxicity via increasing autophagic flux [23]. ZKSCAN3 represents the transcriptional counterpart of TFEB, since it represses the transcription of a number of autophagy-related genes, including Unc-51-like autophagy activating kinase 1 (ULK1). Upon autophagy induction, ZKSCAN3 translocates from the nucleus to the cytoplasm, allowing the transcriptional activation of target genes by TFEB. Interestinly, ZKSCAN3 knockdown is sufficient to induce autophagy, whereas its overexpression can inhibit autophagy [24]. BRD4, an evolutionarily conserved autophagy repressor, inhibits autophagic and lysosomal activities by repressing the expression of autophagy and lysosome genes under nutrient-rich conditions, and derepression of this program contributes to autophagy activation during nutrient deprivation [25].’
8. The figures presented in this review are indeed valuable. This review could be supported by a table showing autophagy gene mutations in the heart, accumulating polypeptide, phenotype and animal model associated with that mutation and if possible human equivalent.
We checked the literature on autophagy gene mutations and their role in the heart. So far only a limited amount of literature is available, which we added to the manuscript at page 7, Lines 220-225:
‘Defective autophagy was reported in HCM caused by Danon disease, Vici syndrome and LEOPARD syndrome. These cardiomyopathies are related to a defect in a gene encoding a protein involved in the autophagy-lysosomal pathway, leading to protein accumulation [64-66]. Interestingly, incomplete mitophagic flux and mitochondrial dysfunction were observed in both in vitro and in vivo models of Danon disease [67]. Furthermore, altered autophagy was also found in HCM directly caused by the mutations in MYBPC3, the most frequently mutated gene in HCM [68].’
9. While elaborating on cardiomyocyte stressors and mitophagy, discussion on the influence of calcium, lipids and exercise would strengthen the review.
As mentioned above, we added information on the role of calcium, lipids and exercise to the manuscript.
10. In the section on AMPK and mitophagy, information is missing on mouse models of AMPK modification and mitophagy, and upstream kinases of AMPK and its impact on autophagy.
We added information on how AMPK regulates mitophagy in mouse models and how upstream kinases of AMPK, such as LKB1 and CaMKK-β, affect autophagy in the text of section 2.2.
We added information on page 4, lines 110-117.
‘As an energy sensor, AMPK is activated during metabolic stress, that either interfere with ATP production (e.g., glucose deprivation, hypoxia, and ischemia) or accelerate ATP consumption (e.g., muscle contraction). Activation of AMPK requires the phosphorylation of its α subunit at Thr172 by two upstream kinases, liver kinase B1 (LKB1) and Ca2+/calmodulin-dependent kinase kinase (CaMKK)-β. Activation of AMPK initiates autophagy, not only through the inhibition of mTOR, but also by direct phosphorylation and, thereby, activation of Ulk1 (Figure 2b) [28]. Chronic AMPK activation restores cardiac function by upregulating autophagy activity in diabetic OVE26 mice (type I diabetes) [29].
11. The minor point is that the figures are not called out much and are not used effectively to guide the reader. Given the complexity of the text and processes being described, labelling up the figures into subsections and calling them out frequently to follow the text would significantly help the reader through this data. Please label the pathways in the figures in alphabetical order and refer to them in the text.
In order to improve guidance to the reader, we labeled the pathways in figure 1 and figure 2 in alphabetical order, and also referred to them in the text at section 2.1 and 2.2.

Reviewer 2 Report
This is a very well written review, which comprehensively describes main findings and current status on the role of authophagy in a number of cardiac disease conditions.
This review would benefit from mentioning/reviewing ischemic heart disease (IHD) with some evidence on atherosclerosis (as far as IHD is concerned) and diabetic cardiomyopathy.
Similarly, a table of main pathways and mechanisms involved in each of the reviewed pathologies would be of a great benefit to the Reader.
Author Response
Reviewer 2
This is a very well written review, which comprehensively describes main findings and current status on the role of authophagy in a number of cardiac disease conditions.
This review would benefit from mentioning/reviewing ischemic heart disease (IHD) with some evidence on atherosclerosis (as far as IHD is concerned) and diabetic cardiomyopathy.
Thank you for your comment. We added the role of autophagy and mitophagy in diabetic cardiomyopathy and atherosclerosis in section 3.2.2 and 3.3.1 and Table 1.
Page 8-9, Lines 243-289:
3.2.2 Diabetic cardiomyopathy
Diabetic cardiomyopathy is characterized by ventricular dysfunction that increases the risk of heart failure and mortality in diabetic patients independent of vascular pathology [75]. There is a strong indication that autophagy is involved in the pathophysiology of diabetic cardiomyopathy, however, the exact role of autophagy in diabetic cardiomyopathy remains controversial [29, 76, 77]. It was reported that autophagic adaptations in diabetic cardiomyopathy differ between type 1 and type 2 diabetes [76]. Moreover, autophagy in the heart is enhanced in type 1 diabetes, but is suppressed in type 2 diabetes. Cardiac damage in streptozotocin (STZ)-induced and OVE26 type 1 diabetic heart is ameliorated in Beclin 1- or ATG16-deficient mice [78]. In high fat diet (HFD)-induced diabetic cardiomyopathy, increased mTORC1 activity contributes to the development of diabetic cardiomyopathy, and mTORC1 inhibition prevents the development of HFD-induced diabetic cardiomyopathy by improving hepatic insulin sensitivity in obesity [79]. Conversely, two studies show that autophagy is suppressed in hearts of STZ-induced diabetic mice and OVE26 type 1 diabetic model mice [29, 78]. One study from Xie et al. reports that suppressed autophagy in hearts of STZ-treated mice and OVE26 type 1 diabetic model mice improve cardiac function by reducing AMPK activity [29]. The other study from Xu et al. shows that diminished autophagy in the same models is an adaptive response that limits cardiac dysfunction in type 1 diabetic heart by increasing mTORC1 activity [78]. Therefore, further research is needed to determine whether autophagy is beneficial or detrimental in diabetes. Interestingly, exercise has beneficial effects on human health, including protection against metabolic disorders such as obesity and diabetes [80]. Interestingly, exercise has beneficial effects on human health, including protection against metabolic disorders such as obesity and diabetes [80]. BCL2 is a crucial regulator of exercise-induced autophagy in vivo, and autophagy induction may contribute to the improved metabolic effects of exercise, indicating that exercise has potential beneficial effects in diabetic cardiomyopathy [81].’
3.3. Ischemic heart disease
3.3.1 Atherosclerosis
Atherosclerosis is a progressive and complex disease that causes the buildup of plaques inside the wall of arteries. Due to technical limitations, investigation on the role of autophagy in plaque formation is not solved yet. Based on existing evidence, triglycerides or other dietary lipids are degraded via autophagy to provide free fatty acid substrates under physiological conditions [82]. However, in the heart, our understanding of the lipid-induced regulation of autophagy is still emerging. In vitro studies in H9C2 cardiomyocytes recapitulate some of the in vivo findings on lipid overload-induced cardiac autophagy [83]. Other studies showed that in human endothelial cells, oxidized LDLs (OxLDLs) trigger the activation of autophagy by upregulating Beclin1 expression, which contributes to phagocytosis of oxLDLs exposed cells [84]. In an ApoE−/− mice study, the importance of autophagy in atherosclerosis progression was further demonstrated by utilizing specific smooth muscle cells (SMC) containing a deletion in Atg7. These SMCs exhibited accelerated atherosclerotic plaque development after 10 weeks of high fat diet [85]. Interestingly, it has been recently reported that PINK1 or Parkin knockdown increases the cytotoxic response of human vascular SMC exposed to OxLDL, whereas PINK1 or Parkin overexpression has cytoprotective effects, suggesting that mitophagy plays a critical role in modulating vascular SMC fate by favoring cell survival or by enhancing apoptosis [86]. Furthermore, in a macrophage-specific ATG5 knockout mice model, the study shows that autophagy becomes dysfunctional in atherosclerosis and its deficiency promotes atherosclerosis in part through inflammasome hyperactivation [87]. However, the relevance of beneficial effects of autophagy in the early stages of atherosclerosis and the detrimental effects of autophagy observed in the late stages of mouse atherosclerotic plaque formation, remains to be further demonstrated in human clinical samples.’
Similarly, a table of main pathways and mechanisms involved in each of the reviewed pathologies would be of a great benefit to the Reader.
We agree and added the main pathways and mechanisms involved in the regulation of autophagy during the various pathologies to table 1.
Round 2
Reviewer 1 Report
Authors have satisfactorily addressed reviewer concerns